# Reinforcement of Calcium Phosphate Cement with Hybrid Silk Fibroin/Kappa-Carrageenan Nanofibers

**DOI:** 10.3390/biomedicines11030850

**Published:** 2023-03-10

**Authors:** Fahimeh Roshanfar, Saeed Hesaraki, Alireza Dolatshahi-Pirouz, Mohsen Saeidi, Sara Leal-Marin, Birgit Glasmacher, Gorka Orive, Sajjad Khan Einipour

**Affiliations:** 1Biomaterials Group, Department of Nanotechnology & Advanced Materials, Materials and Energy Research Center, Karaj 31779-83634, Iran; 2Department of Health Technology, Institute of Biotherapeutic Engineering and Drug Targeting, Center for Intestinal Absorption and Transport of Biopharmaceuticals, Technical University of Denmark, 2800 Kongens Lyngby, Denmark; 3Stem Cell Research Center, Golestan University of Medical Sciences, Gorgan 49341-74515, Iran; 4Institute for Multiphase Processes (IMP), Leibniz University Hannover, 30823 Garbsen, Germany; 5Lower Saxony Centre for Biomedical Engineering, Implant Research and Development (NIFE), 30625 Hannover, Germany; 6NanoBioCel Research Group, School of Pharmacy, University of the Basque Country (UPV/EHU), 01006 Vitoria-Gasteiz, Spain; 7NanoBioCel Research Group, Bioaraba Health Research Institute, 01009 Vitoria-Gasteiz, Spain; 8Biomedical Research Networking Centre in Bioengineering, Biomaterials and Nanomedicine (CIBER-BBN), Institute of Health Carlos III, Av Monforte de Lemos 3-5, 28029 Madrid, Spain; 9University Institute for Regenerative Medicine and Oral Implantology-UIRMI (UPV/EHU-Fundación Eduardo Anitua), 01007 Vitoria-Gasteiz, Spain; 10Singapore Eye Research Institute, The Academia, 20 College Road, Discovery Tower, Singapore 169856, Singapore; 11Department of Tissue Engineering and Regenerative Medicine, School of Medicine, Qom University of Medical Sciences, Qom 37169-93456, Iran

**Keywords:** calcium phosphates cement, nanofiber reinforcement, silk fibroin, mechanical strength, kappa-carrageenan

## Abstract

Calcium phosphate cements (CPCs) offer a promising solution for treating bone defects due to their osteoconductive, injectable, biocompatible, and bone replacement properties. However, their brittle nature restricts their utilization to non-load-bearing applications. In this study, the impact of hybrid silk fibroin (SF) and kappa-carrageenan (k-CG) nanofibers as reinforcements in CPC was investigated. The CPC composite was fabricated by incorporating electrospun nanofibers in 1, 3, and 5% volume fractions. The morphology, mineralization, mechanical properties, setting time, injectability, cell adhesion, and mineralization of the CPC composites were analyzed. The results demonstrated that the addition of the nanofibers improved the CPC mixture, leading to an increase in compressive strength (14.8 ± 0.3 MPa compared to 8.1 ± 0.4 MPa of the unreinforced CPC). Similar improvements were seen in the bending strength and work fracture (WOF). The MC3T3-E1 cell culture experiments indicated that cells attached well to the surfaces of all cement samples and tended to join their adjacent cells. Additionally, the CPC composites showed higher cell mineralization after a culture period of 14 days, indicating that the SF/k-CG combination has potential for applications as a CPC reinforcement and bone cell regeneration promoter.

## 1. Introduction

Over the past decades, various materials have been applied to repair bone defects and deformities. Apatite-forming calcium phosphate cements (CPCs) have emerged as a highly popular option, particularly for the treatment of osteoporosis-related fractures, craniofacial abnormalities, and deformities, as well as in vertebroplasty procedures [1,2,3]. The advantage of CPCs compared to other biomaterials lies in their structural similarity to hydroxyapatite, as well as the mineral component of native bone, which promotes the integration of mineralized tissue and enhances bone repair [4]. Additionally, CPCs possess other desirable qualities, such as osteoconductivity, biocompatibility, moldability, and injectability, making them ideal for minimally invasive surgical procedures. Moreover, they also offer the possibility of loading drugs such as vancomycin, flomoxef sodium, and gentamicin that can accelerate the healing and reconstruction of bone injuries [5,6,7]. Despite these benefits and the commercial and clinical availability of CPCs, there are still some limitations regarding their applications, mainly due to their low mechanical strength and high brittleness. Therefore, their utilization in non-load-bearing bone defects, such as the treatment of certain maxillo-craniofacial defects, is restricted [8]. This can present obstacles, especially in the early stages after implantation, where the low mechanical strength of CPCs can pose a challenge [9,10]. Regardless of these limitations, various strategies have been employed to reduce the brittleness and increase the toughness of CPCs, such as fiber reinforcement [11]. 

Reinforcing CPCs with biofibers is one promising strategy, as it has been previously shown to increase the composite’s fracture toughness, tensile, and flexural strengths, as well as to arrest cracks through crack-arresting mechanisms [10,12]. The low resorption rate of CPCs can also be improved by incorporating biodegradable and biocompatible reinforcing agents [9]. Several polymer fibers, such as polylactic acid/polyglycolic acid [13,14], polyamide [14,15], polyglactin [16], carbon fibers [17], chitosan [18], carbon nanotubes [19], and glass fibers [20], have been employed for this purpose. 

One of the main challenges for the fiber reinforcement of CPCs is the integration of the fibers into the CPC mixture [15]. If they are not properly integrated, the mixture can suffer from decreased injectability and a longer setting time [21]. In a previous study, it was reported that the strength of CPC was enhanced by adding medical sutures made of polylactic-co-glycolic acid (PLGA), whereas they could not be injected because of long fibers and the lack of integrity [1]. Additionally, the fiber reinforcement of ceramic materials enhances fracture resistance but simultaneously reduces the strength of the composite [17]. Some common reinforcement agents, such as carbon nanotubes, have immunogenic responses [22]. Meanwhile, synthetic polymers, such as polylactic acid, can affect the degradation rate, depending on their composition [23]. The key issue is to identify a reinforcement that enables a harmonious blend of the mechanical and biological properties [17]. The addition of nondegradable high-strength fibers can significantly improve the toughness and strength of CPCs, but their low biodegradability prevents the new bone from growing in [24]. With regard to these concerns, the current research on CPC reinforcement materials is focused on finding components that have an appropriate balance between mechanical and biological properties. Regarding the available options for biopolymers, Bombyx mori silk fibroin (SF) fibers possess exceptional biological attributes, including biodegradability, biocompatibility, and the ability to promote the development of new bone tissue [25]. Additionally, these fibers exhibit superior mechanical properties, such as increased tensile strength and toughness, compared to other biological polymers such as gelatin and chitosan [25]. Recently, our group has successfully developed hybrid nanofibers based on SF and kappa-carrageenan (k-CG) via electrospinning [26]. k-CG has been used in drug delivery and tissue engineering due to its gelation properties, forming gels with proteins and drugs, as well as stimulating cell growth [27]. k-CG can also induce the formation of a bone-like apatite layer in the body, as suggested by a previous report [28]. This coincides with the results of our study, where k-CG improved the bioactivity of SF and the osteogenic potential [26].

Following these encouraging results, the aim of this study is to include hybrid SF/k-CG electrospun nanofibers as a reinforcement material for CPC. The morphology, mineralization, mechanical properties, setting time, injectability, cell adhesion, and mineralization were analyzed with different reinforcement volume fractions. To the best of our knowledge, no previous studies have explored the effect of using hybrid SF/k-CG electrospun nanofibers as a reinforcement agent for CPCs to obtain a CPC composite that promotes bone growth with enhanced mechanical properties and future possibilities to be used as sustained time-release drug delivery system.

## 2. Materials and Methods

### 2.1. Chemicals 

Silk cocoons from Bombyx mori were supplied by the Golestan Silk Research Center. Kappa-carrageenan (k-CG), 1,1,1,3,3,3-Hexafluoroisopropanol (HFIP), and chloroform (CHCl_3_) were purchased from Sigma-Aldrich (St. Louis, MO, USA). Calcium carbonate (CaCO_3_), dicalcium phosphate dihydrate (CaHPO_4_·2H_2_O), absolute ethanol 99.7%, genipin (GP), and isopropanol were bought from Merck (Wetzlar, Germany). Furthermore, fetal bovine serum (FBS) and cell culture materials were purchased from Nemooneeh Vasegh Company (Gorgan, Iran).

### 2.2. Synthesis of CPC Composites

#### 2.2.1. Calcium Phosphate Cement

The calcium phosphate cement (CPC) powder was formed from a mixture of tetracalcium phosphate (TTCP; Ca_4_(PO_4_)_2_O) and anhydrous dicalcium phosphate (DCPA; CaHPO_4_). TTCP was made through a solid-state reaction between CaCO_3_ and CaHPO_4_·2H_2_O (DCPD), according to the method described in previous studies [29]. Briefly, 1 mol of DCPD and 1 mol of CaCO_3_ were combined, followed by milling for 1 h and heating for 6 h at 1500 °C. In the end, the product was crushed and milled in a planetary mill to an average particle size of 13 µm. Then, a 1:1 molar combination of TTCP and DCPA with an average particle size of 7 µm was used as the cement’s solid phase.

#### 2.2.2. Electrospinning Silk Fibroin/k-CG

The solution preparation and electrospinning parameters were based on our published protocol for hybrid SF/k-CG [26]. Briefly, a freeze-dried SF sponge was put in an 8:2 (*v*/*v*) HFIP/CHCl_3_ solution and stirred for about 3 h to prepare a 12% (*w*/*v*) SF solution. The SF solution was then mixed in a shaker with 1 mg/mL of k-CG powder for 24 h to dissolve. The solution was transferred into a 10 mL syringe with a 21-gauge needle for electrospinning (Nanoazma Co., Tehran, Iran). A syringe pump, with the syringe inside of it, was positioned vertically with a distance of 12 cm from an aluminum collecting plate (dimensions 12 × 10 cm^2^). Regarding the electric field, a voltage of 20 kV was applied between the aluminum collecting plate and the needle tip. A flow of 0.3 mL/h was utilized to dispense the polymer solution from the syringe. Finally, 1% (*w*/*v*) GP was used to crosslink the nanofibers accumulated on the aluminum foil surface. To do this, GP powder was first dissolved in distilled water and stirred for 30 min to create a homogeneous solution. Afterwards, the electrospun fibers submerged 24 h in the GP solution.

#### 2.2.3. Composite CPC

Crosslinked SF/k-CG fibrous membranes were cut into 3 × 3 mm^2^ pieces to prepare the fiber-reinforced CPC composites. The size was selected since previous studies showed that 3 mm fibers, incorporated into the CPC paste, were optimal for preparing an injectable CPC [9,30]. The CPC powder described in Section 2.2.1 was mixed with distilled water to create a paste with a 3 g/mL powder-to-liquid ratio. Using a spatula, the cut electrospun SF/k-CG bundles were manually mixed with cement paste at varied volume ratios of 1, 3, and 5% (based on the whole volume of the powder and liquid). The resulting composites were named CPC-1, CPC-3, and CPC-5, respectively. The CPC without nanofibers (CPC-0) was considered as the control sample. 

### 2.3. Mineralization CPC Composites

As first approach for the possible bone-like apatite phase formation, the disc-shaped CPC composites (Ø 10 mm × 3 mm) were immersed in 20 mL of simulated biological fluid (SBF) for 7 days at 37 °C; the solution was changed every day. SBF solution was prepared following the protocol described in a previous study [25]. In brief, NaHCO_3_ (0.350 g), NaCl (7.996 g), KCl (0.224 g), MgCl_2_·6H_2_O (0.305 g), K_2_HPO_4_·3H_2_O (0.228 g), Na_2_SO_4_ (0.071 g), and CaCl_2_ (0.278 g) were dissolved in deionized water and buffered to obtain a pH value of 7.4 at 37 °C.

### 2.4. Characterization

#### 2.4.1. Morphological Analysis

The morphology of electrospun nanofibers and the prepared composite cement cross-sections were examined using SEM (TESCAN MIRA3). For this purpose, the samples were first mounted on an aluminum stub to be sputter-coated with a layer of gold (Leica EM SCD005, Wetzlar, Germany) for 1 min. When sputtering the samples with gold, the value of the vacuum was 10 Pa. To observe the samples’ morphology, high-resolution SEM with a solid-state secondary ion detector and an acceleration voltage of 30 kV was employed. The average diameter of the electrospun fibers was measured by detecting 50 fibers randomly from SEM images and analyzing them by using ImageJ^®^ 1.53g software (NIH, Bethesda, MD, USA). Additionally, the surfaces of the CPC composites were observed by SEM after cell seeding, following the preparation described in Section 2.4.6. 

#### 2.4.2. X-ray Diffraction (XRD) Analysis

An automated X-ray diffractometer (Philips PW 3710) was used to determine the XRD pattern of the CPC composites after 24 h of incubation and also after immersion in SBF solution for 7 days. The main purpose was to identify crystalline phases of hydroxyapatite after mineralization. The composites and CPC powder as references were measured by triplicate from diffraction angles of 2θ = 20–40° at a scan rate of 0.005°/s using a CuKa radiation source (wavelength = 0.154 nm).

#### 2.4.3. Setting Time 

The setting time of the CPC composites was measured utilizing a Gillmore apparatus at room temperature (ASTM C266-89) [4]. The initial and final setting times were determined by needles with a weight of 113.4 g and 453.6 g, respectively. The time was recorded when the needle could not leave a visible print on the surfaces of the sample. The composites were tested in triplicate.

#### 2.4.4. Mechanical Tests

The mechanical properties of the CPC composite cements were tested using a universal mechanical testing machine (Zwick/Roell-HCR 25/400). To measure the compressive strength, the cylindrical samples (Ø 6 mm × 12 mm) were stored for 24 h in an incubator (37 °C and 99% humidity) and were evaluated at a cross-head speed of 1 mm/min. The bending strength was measured in cuboid-like samples (3 × 4 × 40 mm^3^) at a cross-head speed of 1 mm/min by three-point bending methods using an outer span of 20 mm. The bending strength and work of fracture (WOF) of the samples were calculated based on the load–displacement curve and according to Equations (1) and (2) [4,9]. All mechanical tests were performed in triplicate.
(1)$$Bending\;Strength\;:S=3\times F_{max}\frac{L}{2\times b\times d^{2}}$$
where *F_max_* is the ultimate load, *L* is the bending span, *b* is the width of the sample, and *d* is the thickness.
(2)$$WOF=\frac{A}{b\times d}$$
where *A* represents the area under the load–displacement curve.

#### 2.4.5. Injectability

The injectability of the CPCs was performed by a Zwick/Roell-HCR 25/400 mechanical device at a cross-head speed of 5 mm/min [1]. Two minutes after mixing the cement components, the as-prepared paste was transferred into a 10 mL syringe (inner tip diameter of 800 μm). The paste started to come out of the reservoir by a force that was applied vertically to the end of the syringe. Load–displacement curves were achieved for all four formulations.

#### 2.4.6. Cell Seeding and Morphology

To assess the biocompatibility and cell adhesion on the CPC composites, the disc-shaped samples (Ø 10 mm × 3 mm) were seeded with M3T3-E1 cells. First, the discs were sterilized with 75% ethanol and placed on 24-cell culture well plates, followed by cell seeding at a density of 3 × 10^4^ cells/disc in 200 µL medium. Afterwards, the discs were placed in an incubator at 37 °C with 5% CO_2_ for 4 h to allow the cells to adhere to the surface. Then, each well was filled with 2 mL medium and incubated for 24 h. The cell culture medium was DMEM containing 10% FBS and 1% penicillin–streptomycin. After 24 h, the discs were rinsed with PBS and cells fixed with 4% (*v*/*v*) glutaraldehyde for 90 min, rinsed with deionized water, and dehydrated with graded ethanol solution. Once dehydrated, the samples were dried in the air and coated for SEM, as described in Section 2.4.1.

#### 2.4.7. Cell Mineralization 

Alizarin Red staining was utilized to assess the calcium content of the composite CPC cement after cell mineralization. Cell seeding was performed as previously described in Section 2.4.6—in this case, with cell culture medium for osteogenic differentiation (DMEM supplemented with 10% FBS, 1% penicillin–streptomycin, 0.1 g/mL dexamethasone, 50 g/mL ascorbic acid, and 10 mM glycerophosphate). Samples were incubated for 14 days, followed by fixation with 4% (*w*/*v*) paraformaldehyde for 30 min. Then, 1 mL of Alizarin Red solution was added to each well and incubated for 1 h at room temperature. After incubation, the samples were washed with deionized water, dried at room temperature, and analyzed under an inverted microscope (Olympus Microscope, Tokyo, Japan) for a qualitative analysis. In order to quantify the calcium concentration, each stained sample was immersed in 1 mL of 10% cetylpyridinium chloride solution for 2 h to dissolve the calcium ions. The solution’s absorbance was determined at 540 nm in an ELISA microplate reader (Stat Fax-4700). 

#### 2.4.8. Statistical Analysis 

All the experiments were performed at least three times. SPSS 17.0 software was used for data analysis, and the results were presented as the mean ± SD. To compare the results, ANOVA (one-way analysis of variance) was applied, and for significance, a threshold of *p* < 0.05 was set in the statistical calculations.

## 3. Results and Discussion

### 3.1. SEM Observations of SF/k-CG Fibers

The hybrid fibers of SF/k-CG obtained by electrospinning are presented in Figure 1. The fibers range between nano- to micrometric size, with a mean diameter of 450 ± 9 nm. Thicker fibers could be seen as agglomerations of thinner fibers bound together before complete solvent evaporation during the electrospinning process. In general, fibers present a homogenous morphology with a smooth surface and without pores or beads, indicating an optimal mix between the two components (SF/k-CG). Furthermore, the fibers are randomly aligned due to the type of collector employed.

### 3.2. Morphology and Microstructure of CPC Composites 

To study the interior of the specimens, cross-sections of the fractured pieces were examined by SEM. Figure 2 shows the cross-sectional surfaces of typical CPCs with SF/k-CG nanofiber volume fractions of (A) 0%, (B) 1%, (C) 3%, and (D) 5%. The control sample CPC-0 without fibers showed a typical grain morphology for calcium phosphate cements [20]. The grains are reduced proportionally to the increase in the nanofiber volume fraction, as presented in CPC-1, CPC-3, and CPC-5. The fibers kept the random alignment and showed a filling of CPC paste between them, especially for CPC-5. Figure 2 also demonstrates that the fibers are evenly distributed throughout the composite samples of the cement matrix. The fibers and CPC paste seem to have combined together successfully, with the CPC separating the nearby fibers and the fibers possessing various orientations. This can be due to the modification of the hydrophilic properties of SF with k-CG polysaccharide, which then blends better with the hydrophilic surface of calcium-phosphate cement [16]. In addition, as expected, more fibers were observed on the surface at an increase in nanofibers from 1 to 5% by volume.

### 3.3. XRD 

Figure 3A,B represent the XRD spectra for the different types of cement after incubation for 24 h and 7 days of immersion in SBF solution, respectively. As shown in Figure 3A, in all the cement samples, the peaks related to the apatite phase appeared at 26° and 32° (JCPDS card No. 09-0432) as a product of the setting reaction and are presented along with the peaks related to TTCP at 29.4° and 29.8° (JCPDS card No. 0232-11) and DCPA at 21° and 31° (JCPDS card No. 9-77) as the initial materials (reactants) [1,31]. Progress in the formation of an apatite product after soaking in SBF for 7 days is illustrated in Figure 3B. In all types of cement, the initial reactive phases almost disappeared, and apatite is the only phase present in the cement. It seems that the presence of SF/k-CG hybrid nanofibers in composite cement did not cause a delay in the rate of apatite phase formation. Additionally, an increased rate of apatite phase formation in the composite cements could be observed compared to the control sample. This can be due to the presence of k-CG in the fiber structure and the influential role of its sulfonate groups (-SO_3_H) in absorbing calcium ions from the environment and accelerating the process of hydroxyapatite formation [25]. In fact, -SO_3_H groups are suitable locations for germinating apatite crystals and encourage the precipitation of hydroxyapatite by forming a calcium sulfate complex [26].

### 3.4. Setting Time

The CPC setting time is important in orthopedic applications to guarantee the defect coverage and enough strength within the material to withstand a force [32]. Figure 4A shows the different CPC composites’ initial and final setting times. Compared to the control sample, all samples generally have faster setting times after reinforcing the cement formulation with nanofibers. For sample CPC-3, containing 3% SF/k-CG hybrid nanofibers, the initial and final setting times shortened significantly (*p* < 0.001) to 5 and 20 min compared to fiber-free cement (CPC-0), which needed more time to set (it took 10 and 35 min for CPC-0 regarding the initial and final setting times, respectively). This time reduction was observed less noticeably when the volume fraction of nanofibers rose to 5%. In general, including hydrophilic k-CG into the fiber structure reduces the liquid/powder ratio in cement samples containing hybrid fibers, which can shorten the setting time in composite samples [15]. These outcomes are in line with the findings of earlier investigations, which showed that the presence of polycaprolactone (PCL) fibers and its surface modification with polylactic acid (PLLA) could accelerate the setting of calcium phosphate in cements [9]. In another study, it was reported that the hydrophilicity of electrospun bioactive glass nanofibers could reduce the water ratio of the paste and eventually improved the cement setting [31]. Additionally, the interlocking of the apatite crystals deposited from the cement paste causes the initial setting reaction in calcium-phosphate apatite cements.

### 3.5. Mechanical Strength

The mechanical characteristics of the different CPC composites were evaluated compared to the control (CPC-0) and are shown in Figure 4B–D. The compressive strength, bending strength, and WOF values for CPC-1 and CPC-3 were all significantly greater than CPC-0. CPC-3 showed an increase in compressive strength (14.8 ± 0.3 MPa) in comparison to CPC-0 (8.1 ± 0.4 MPa). The values without reinforcement are similar to the values reported in other CPC studies, while the reinforcement with SF/k-CG presented a higher compressive strength in comparison with different percentages of polylactic acid (PLA) fibers, which maximum values were around 10 MPa [20]. A similar trend of increasing values was observed in the results obtained from the bending strength and WOF. The WOF value of 6 kJ/m^2^ for CPC-3 is similar to the one in the literature for reinforcing carbon fibers, although, for this material, the compressive strengths achieved are considerably higher (59 MPa) compared to SF/k-CG [17].

All mechanical properties for the composite cement containing 5% SF/k-CG hybrid nanofibers (CPC-5) were significantly lower than those for 3% SF/k-CG hybrid nanofibers (CPC-3). It could also be shown that adding more reinforcing nanofibers not only improved the mechanical properties of CPC but also acted as defect sites in the microstructures of the composite cements and generally reduced all the mechanical properties. It is crucial to note that the mechanical characteristics of the composite cements are typically better than those of unreinforced sample (control sample). According to reports, the mechanical properties of unidirectional fiber-reinforced composites cements show a continuous enhancement with an increase in the fiber volume fraction. In contrast, random fiber composites’ mechanical strength increases until the fiber volume fraction reaches a specific maximum value and then begins to decline [1,4,21]. The mechanical results obtained by a previous study indicated that excessive proportions of bioglass fibers acted as defect sites in the microstructure of cement and led to the formation of voids between the bioglass fibers and the cement and finally resulted in deterioration of the mechanical properties [31].

The addition of nanofibers to the cement foundation improved the mechanical properties, as demonstrated by the increase in the WOF values of the composite samples compared to the control. The flexible nature of the polymer fibers contributes to the enhanced flexibility by allowing stretching or deformation through frictional pull-out, thereby increasing the energy consumption of the cement [8,23].

### 3.6. Injectability

Figure 5 provides a detailed analysis of the load–displacement curves of CPC-0 and the CPC composites, which offer valuable insights into the injectability of these materials. As the curves clearly demonstrate, all of the samples experience a dramatic increase in force at the onset of the injection phase. This is a result of the paste’s resistance to friction along the walls of the syringe. In the case of CPC-0, the injection process is smooth and uniform from point A1 to A2, with a relatively constant force. The force required for the injection of CPC-1 remains constant from point B1 to B2, but there is a noticeable increase from B2 to B3. This increase in force could be due to the filter-pressing phenomenon, which occurs when there is a separation of the solid and liquid phases in the cements. The presence of nanofibers containing k-CG, which possess hydrophilic properties, causes water absorption, which, in turn, reduces the liquid-to-powder ratio in the cement composition and leads to an increase in the force required to eject the cement from the syringe [1,31]. The injectability of CPC-3 and CPC-5 was impaired, requiring an increased force at points C2 and D2, respectively, to clear the residual paste from the syringe. In order to achieve a successful clinical application of CPC composites, it is essential to conduct further research with the aim of reducing the force required for injectability while preserving the mechanical properties of the material. This will require a careful balance between optimizing the injectability and maintaining the mechanical strength of the composites.

### 3.7. Cell Morphology

Figure 6 presents a visual representation of the morphological characteristics of MC3T3-E1 cells after 24-h exposure on the surface of the four different types of CPC composites (CPC-0, CPC-1, CPC-3, and CPC-5). The SEM images provide a clear illustration of the cells’ attachment to the sample surface, which is achieved through the extension of their membranes using lamellipodia and pseudopodia. The rough surface texture of the samples and the interaction of the cells’ appendages with them are the key factors contributing to the strong adhesion of the cells. Furthermore, the cells’ expansive state on the sample surface indicates a positive interaction and suggests that all four cement samples exhibit high biocompatibility [25,26,33]. These findings provide evidence of the potential applications of these CPC composites in the field of biomedical engineering, particularly in tissue regeneration.

### 3.8. Cell Mineralization

As shown by Alizarin Red staining, represented in Figure 7A, MC3T3-E1 cells exhibited mineral calcium deposits after 14 days. The quantitative analysis of Alizarin Red staining in Figure 7B revealed that the CPC composites had higher mineral deposits than CPC-0. The effect of mineralization improved as the presence of SF/k-CG hybrid nanofibers increased. Additionally, the CPC-3 and CPC-5 mineral depositions were considerably (*p* < 0.05) higher than the control sample. This is in accordance with a previous study suggesting CG promotes osteogenic differentiation in bone scaffolds [26]. In fact, the existence of sulfonate groups (-SO_3_H) in CG promotes the uptake of Ca^2+^, which leads to calcification through nucleation and growth. In mineralized ECM, Alizarin Red can bind to Ca^2+^, leading to bright red stains. However, it would be necessary to use other quantitative methods to exactly determine the mineralization values achieved in comparison to healthy bone formation.

## 4. Conclusions

In this study, hybrid SF/k-CG fibers as a novel reinforcement material in CPCs were introduced. The study examined the effects of three different volume fractions (1%, 3%, and 5%) on various parameters, including morphology, mechanical properties, setting time, injectability, cell adhesion, and cell mineralization. The findings indicated that samples with 3% nanofibers (CPC-3) exhibited a desirable balance between biological and mechanical properties. Higher fiber volumes (5%) had an adverse effect on the mechanical properties, although increased cell mineralization. In vitro cell culture studies corroborated the absence of cytotoxicity, along with notable cell attachment and mineralization, for all composite samples. To summarize, this study provides valuable insights into the use of SF/k-CG fibers as a reinforcement material in CPCs and lays the foundation for later in vivo investigations and clinical bone healing applications to optimize their properties and applications.

## Figures and Tables

**Figure 1 biomedicines-11-00850-f001:**
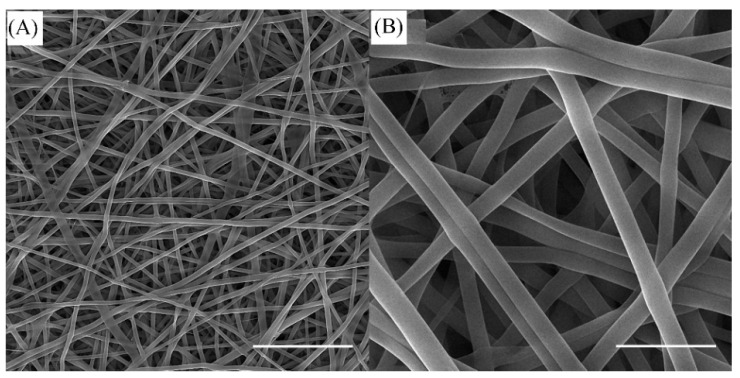
SEM images of hybrid electrospun SF/k-CG nanofibers. Scale bar is 10 µm (**A**) and 2 µm (**B**), respectively.

**Figure 2 biomedicines-11-00850-f002:**
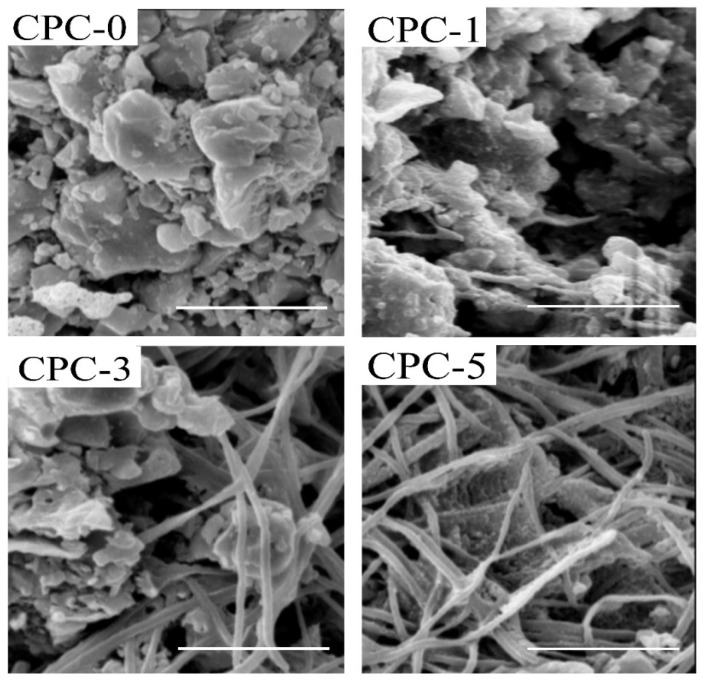
SEM images of cross-sectional surfaces of CPC composites. CPC appeared to be bonded onto the fiber surfaces. The electrospun SF/k-CG nanofibers were mixed well within the CPC paste. The scale bar is 10 µm.

**Figure 3 biomedicines-11-00850-f003:**
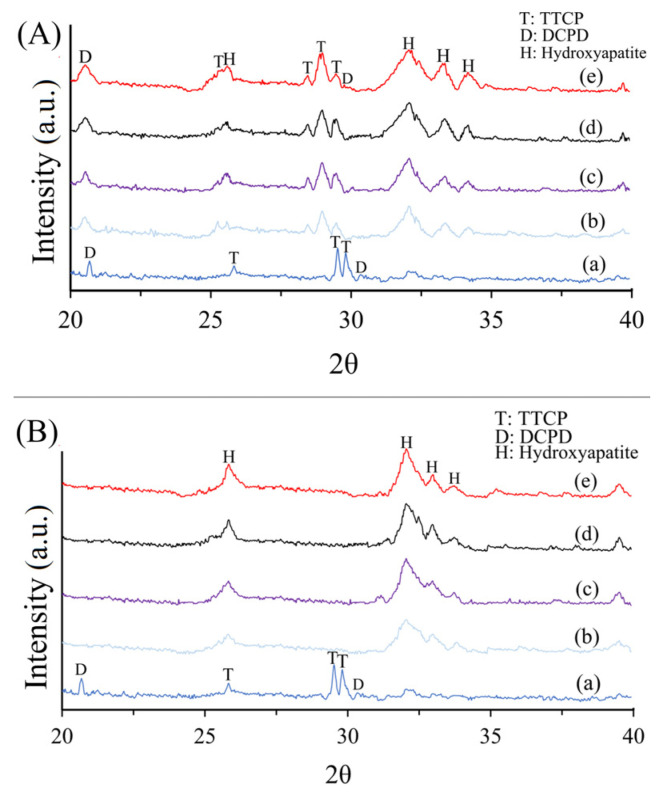
XRD patterns of different samples (**A**) after incubation for 24 h (**B**) after 7 days of immersion in SBF solution. (a) Cement powders, (b) CPC-0, (c) CPC-1, (d) CPC-3, and (e) CPC-5.

**Figure 4 biomedicines-11-00850-f004:**
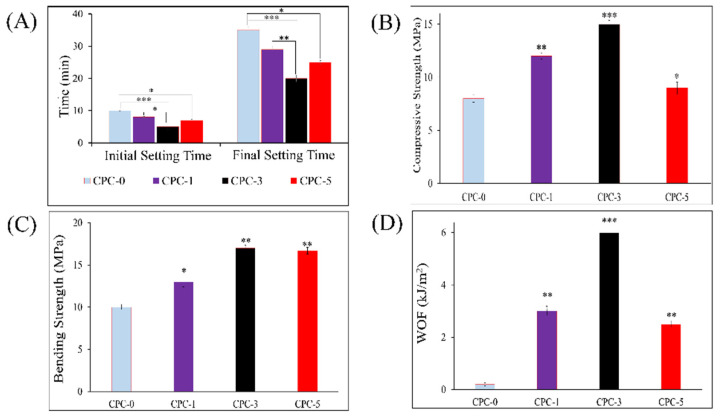
(**A**) The setting times of the different samples, (**B**) compressive strength, (**C**) bending strength, and (**D**) work-of-fracture values. (***, **, and * show significant differences between each group and the control sample (CPC-0) at *p* < 0.001, *p* < 0.01, and *p* < 0.05, respectively).

**Figure 5 biomedicines-11-00850-f005:**
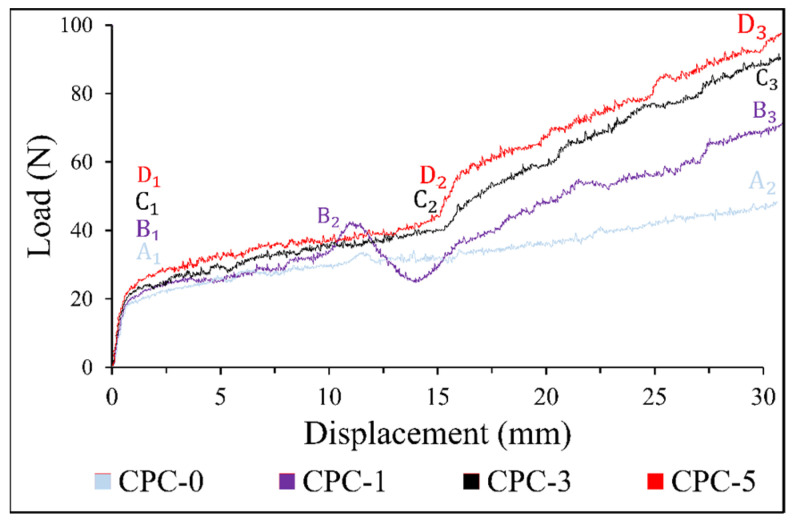
The load–displacement curves of the different cement samples.

**Figure 6 biomedicines-11-00850-f006:**
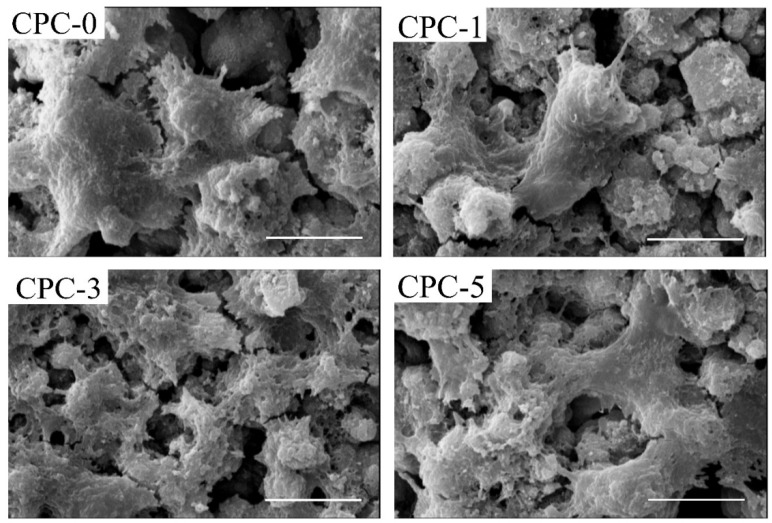
SEM images of the initial attachment of MC3T3-E1 cells on the cement samples. The scale bar is 10 µm.

**Figure 7 biomedicines-11-00850-f007:**
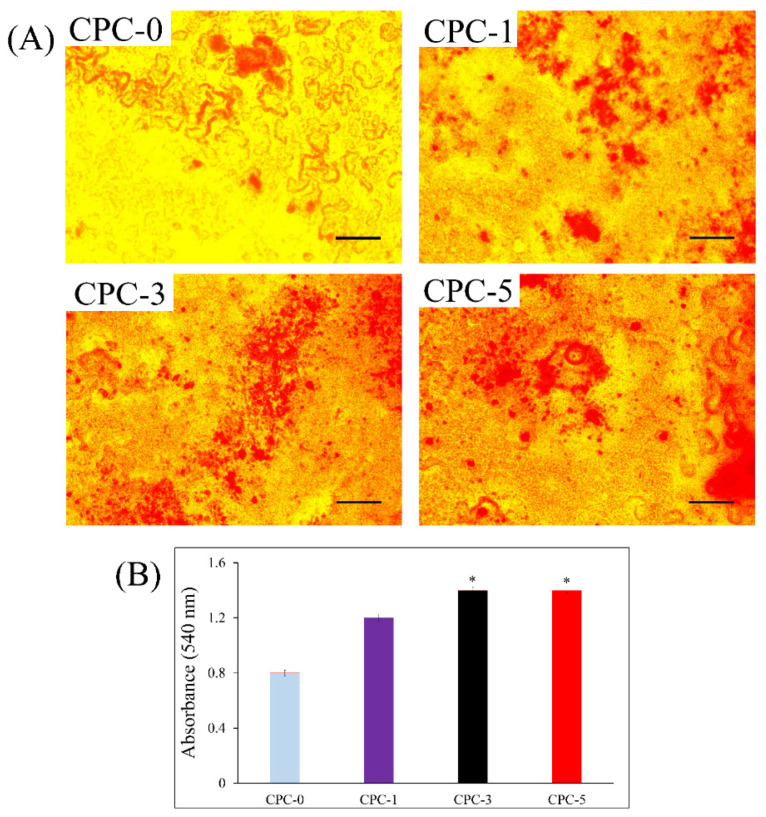
(**A**) Alizarin Red staining of MC3T3-E1 seeded on cement samples on day 14. The scale bar is 200 µm; (**B**) quantitative evaluation of the mineral deposition of the calcium content of MC3T3-E1 cultured on CPCs (* *p* < 0.05).

## Data Availability

Not applicable.

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
