# Peer review of "Reinforcement of Calcium Phosphate Cement with Hybrid Silk Fibroin/Kappa-Carrageenan Nanofibers"

_biomedicines, 2023, doi:10.3390/biomedicines11030850_

Round 1
Reviewer 1 Report
The introduction fails the present the state of knowledge in the RECENT literature concerning the subject and study area of this manuscript and fails to identify a gap that this manuscript proposes to contribute. Please revise.
Please sharpen the description of the novelty factor in the "in this work" section of the introduction. What exactly was done in this study for the first time?
As presented, the title and the manuscript in terms of “mechanical reinforcement” are not properly related.
The Statistical analysis refers only to triplicate samples? Please add more information.
In current form, the level of section is weak to moderate and the manuscript seems to be only an enumeration of information/ obtained results. This issue should be corrected by highlighting the main findings, shortening the information presented, while keeping only the significant results and compare them with similar studies (please apply) in a more "technical" way. Indeed, there is a lot of work and results, but in this form, the manuscript is difficult to read and to remark the most important findings.
Please improve the discussion of biological evaluation in all sections (abstract, results and discussion, conclusions) of the manuscript and reconsider the title, objective and conclusions of the current study.
The Conclusions section should be rewritten/ shorten in order to highlight the following aspects: (a) the problem statement addressed in the paper; (b) summarize overall findings – numerical form and (c) the key takeaways from your paper.
Specific comments: What is “l” in line 136? Figure 2. - Intensity (a.u).
Finally, I believe that the work is not suitable for publication in this form and requires large revision.
Reviewer 2 Report
The analyzed article addresses an interesting and current topic. It is well and logically presented, it meets the requirements of a specialized study in the field of innovative materials. My suggestions and proposals for improvement for a future publication are the following:
1. The Introduction section must be completed with the following:
- Do the incorporation of Hybrid Silk Fibroin/kappa-Carrageenan fibers for Mechanical Reinforcement of Calcium-Phosphate Cement, has any inconvenient, drawback? If so, please detail here.
- The purpose (the aim) of the studies presented in this article must be the subject of the last paragraphs of the Introduction section.
2. The Conclusion section does not reload the aims/scopes of the studies, but instead in this part the authors will present more precisely the grade of fulfilment of those objectives ....
3. Please indicate the next studies that need to be continued at this research theme, also in the Conclusion part.
4. Recheck the References list, to agree with the requirement from the Authors Guide....
5. The Resolution for the diagrams and figures to be homogenized.
Round 2
Reviewer 1 Report
This manuscript is ready for publication in Biomedicines, after English proofreading. They have addressed all my concerns with great satisfaction
Reviewer 2 Report
The manuscript has been properly improved, and the authors have satisfactorily responded to all suggestions and recommendations.